# Deep set autoencoders for anomaly detection in particle physics

**Bryan Ostdiek**

Department of Physics, Harvard University, Cambridge, MA 02138, USA
The NSF AI Institute for Artificial Intelligence and Fundamental Interactions

bostdiek@g.harvard.edu

## Abstract

There is an increased interest in model agnostic search strategies for physics beyond the standard model at the Large Hadron Collider. We introduce a Deep Set Variational Autoencoder and present results on the Dark Machines Anomaly Score Challenge. We find that the method attains the best anomaly detection ability when there is no decoding step for the network, and the anomaly score is based solely on the representation within the encoded latent space. This method was one of the top-performing models in the Dark Machines Challenge, both for the open data sets as well as the blinded data sets.

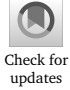

## 1 Introduction

Despite extensive searches for new physics, the experiments at the Large Hadron Collider (LHC) have not yet found physics Beyond the Standard Model (BSM). While there are many

theoretically motivated BSM models, the lack of significant evidence for these models motivates studies of broad, model-agnostic searches. The rise of modern machine learning tools has generated many such data driven methods to search for new physics without a specific model, see Ref. [1] for a recent review of anomaly detection or unsupervised methods.

We break the types of anomaly detection techniques into two categories. The first is methods that use density estimation or look for bumps on smooth curves [2–24]. The recent LHC Olympics [17] introduced four data sets of simulated hadronic LHC events. Many teams submitted results for new methods looking for hidden resonances in the data. The second category of anomaly detection searches for individual events that look different than what is expected from the standard model (also known as out-of-distribution events) [25–54]. The Dark Machines Anomaly Score Challenge [50] studied many methods to flag individual events as anomalous after training on a simulated set of standard model background events. This work was a contributed method to the Dark Machines challenge.

In this paper, we introduce the Deep Set $\beta$ Variational Autoencoder for anomaly detection at the LHC. It is built upon the principle that the reconstructed objects at the LHC can be viewed as sets of particles, or a point cloud. In the mathematical sense, a set is a collection of objects; in our case, the objects are the reconstructed particles. We build upon the particle flow deep set networks introduced in Ref. [55] for supervised classification.[1] The particle flow networks were built on two functions represented as neural networks. The first function, called $\Phi$, operates on each particle within the set in parallel, mapping from the number of input features per particle (four momentum, charge, particle id, etc) to a defined latent output dimension. The outputs for each particle are then summed along the number of particles to make the latent representation of the event permutation invariant. The second function, called $F$, maps from the latent space to the final answer, which for Ref. [55] was a number between 0 and 1 for quark-jet versus gluon-jet discrimination. To implement this method for anomaly detection, we adapt an autoencoder approach, where the original data is compressed to a small latent space and then remapped back to reconstruct the input data; anomalous events are expected to have larger discrepancies between the input and output than the in-distribution data which is used to train the network. Thus, after constructing a permutation invariant latent space, we then use $F$ to generate a new set. To compare the input and output sets, we use the Chamfer loss, which is a method to measure the distance between sets [58].

To force a structure on the latent space, we add a variational step, such that the output of $\Phi$ controls the parameters of a random draw for the representation in latent space. An extra term is added to the loss function which drives the latent representation towards a multidimensional normal distribution. The parameter $\beta$ will control the relative importance of the set reconstruction loss and the latent representation loss,

$$\text{loss} = \beta \times (\text{latent representation loss}) + (1 - \beta) \times (\text{set reconstruction loss}), \qquad (1)$$

with $\beta = 0$ focusing only on the reconstruction loss and $\beta = 1$ focusing only on the latent representation. When evaluating events in the test set, we use the loss as the metric to determine how anomalous the event is, we expect lower loss for SM events and larger loss for BSM events. Unexpectedly, we find that $\beta = 1$ offers the best anomaly detection performance for this method. In fact, our networks with $\beta = 1$ were among the top methods submitted to the Dark Machines challenge [50]. As $\beta = 1$ does not include the reconstruction loss, there is actually no need for the decoder part of the network, and the method is only trying to map the set to a Gaussian latent space. The other top methods of the challenge used density estimation techniques combined with either a variational autoencoder with $\beta = 1$ or a Deep Support Vector Data Description (Deep SVDD) [59] model. The Deep SVDD models work by trying to

---

[1]Graphs have also been used in association with the point cloud representation of the data [56]. Similarly, deep set networks have been studied in the context of self-supervision in Ref. [57].

map the input data to a single fixed vector. The SM background events are expected to get mapped very close to the vector, while BSM events will be further away. Thus, all of the top models in the Dark Machines challenge used methods which either map the input data to a fixed Gaussian distribution or a fixed vector. It is unclear why these so called "fixed target" methods are outperforming models which reconstruct the input.

The paper is organized as follows. In Sec. 2, we briefly review the Dark Machines datasets used for this study. We introduce our network architecture and training details in Sec. 3. The networks are applied to independent test sets in Sec. 4. Conclusions are presented in Sec. 5.

## 2 The Datasets

The data for this work come from the Dark Machines Anomaly Score Challenge [50]. A large dataset of over 1 billion simulated proton-proton collisions with a center of mass energy of $\sqrt{S} = 13$ TeV was generated [60]. This set of events comes from 26 different standard model (SM) processes. From these events, four potential signal regions are examined.

- **Channel 1:** is focused on events with a lot of hadronic activity and missing energy. The selection requirements are

$$H_T \geq 600 \text{ GeV}, \quad E_T^{\text{miss}} \geq 200 \text{ GeV}, \quad E_T^{\text{miss}}/H_T \geq 0.2,$$

  along with at least four jets (including $b$-jets) with $p_T \geq 50$ GeV including at least one with $p_T \geq 200$ GeV. There are 214 000 SM events in this channel.

- **Channel 2:** focuses on leptonic-events. There are two sub channels.

    - **a:** contains at least 3 leptons (muons/electrons) with $p_T \geq 15$ GeV and at least 50 GeV of $E_T^{\text{miss}}$. There are only 20 000 SM events in this channel.
    - **b**: removes the necessity of one of the leptons and replaces it with hadronic activity. The requirements are at least two leptons with $p_T \geq 15$ GeV, $E_T^{\text{miss}} \geq 50$ GeV, and $H_T \geq 50$ GeV. There are 340 000 SM events in this channel.

- **Channel 3:** is more inclusive and contains events which have $H_T \geq 600$ GeV and $E_T^{\text{miss}} \geq 100$ GeV. This channel contains 8 500 000 background SM events.

In addition to the SM events, events were generated for 11 different BSM signals for a total of 18 mass spectra. The BSM models contain a dark matter particle and come from general classes of $Z'$, R parity conserving supersymmetry, and R parity violating supersymmetry (RPV). While the BSM events for each signal may appear in multiple channels, for the most part we focus on anomaly detection within each channel separately.

The events for each channel are available on Zenodo.[2] The data consists of the reconstructed physics objects: non-$b$-tagged jets, $b$-tagged jets, $e^+$, $e^-$, $\mu^+$, $\mu^-$, and $\gamma$. For each object in the event, the four-momentum and object type are available. In addition, the missing transverse momentum and its azimuthal direction are recorded for each event. There were a maximum of 20 objects in each event.

The task of the challenge was to train an anomaly detection method on the SM background events (leaving the final 10% for testing). For any given event, we produce an "anomaly score" which denotes how close or far away the event is from the SM expectation.

---

[2]https://zenodo.org/record/3961917 [61]

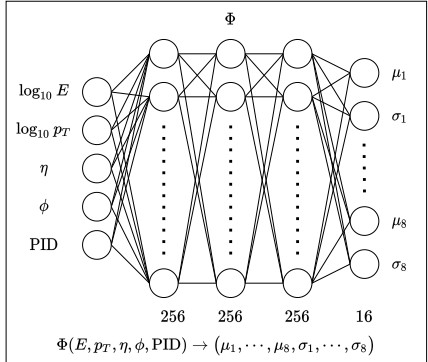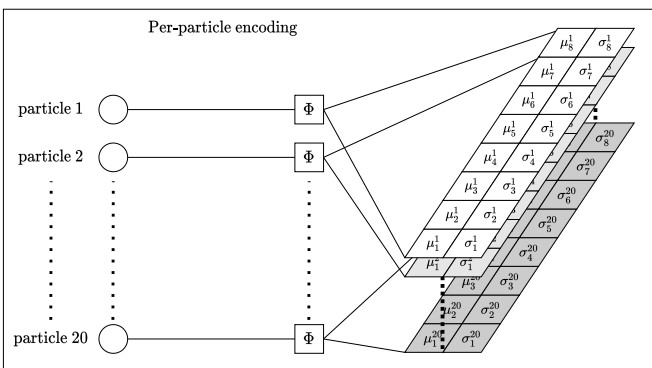

Figure 1: The left panel displays the network $\Phi$ which takes as input the four-vector and particle identification of a particle and maps it into the latent space. The right panel shows that how this is done for each particle in the event. We use a featurewise sort pooling layer to combine the per-particle latent space into an event-level representation as shown in Eq. (2).

## 3 Methodology

### 3.1 The Network

The main idea behind our method is that at the outgoing particles in collider experiments can be thought of as a collection of four-vectors. There is no intrinsic ordering, although they are often sorted by the magnitude of the transverse momentum. Using a network architecture which respects the permutation symmetry is more natural and could lead to improved results. In Ref. [62], it was shown that "Deep Sets" with permutation-invariant functions of variable-length inputs can be parameterized in a fully general way. This idea was introduced to the High Energy Physics community in Ref. [55], where the operations were generalized to include infrared and collinear (IRC) safety. Their methods outperformed other state-of-the art classifiers, and they found a slight improvement if the operations were not restricted to be IRC safe.

As an unsupervised learning task, we modify the deep sets paradigm to include an auto encoding structure, following the example of Ref. [58]. As in [55], we map each particle to the latent space using a function $\Phi$. The functional form of $\Phi$ is a four layer neural network which has inputs of $(\log_{10} E, \log_{10} p_T, \eta, \phi, \text{ID})$. There are then three hidden layers with 264 nodes each, using ReLU activation functions. The final layer of $\Phi$ contains 16 nodes, containing the mean ($\mu_i$) and variance ($\sigma_i$) for each of the eight ($i$) latent dimensions. This is depicted in left panel of Fig. 1. The right panel illustrates how the same function is applied to each particle within the event, leading to many latent representations.

One possible way of combining the per-particle latent spaces into an event-level latent representation is to sum the individual components [58]. However, Ref. [63] showed that the performance can be improved by sorting each feature (for instance all of the $\mu_1$) along all of the particles followed by a learned mapping from the sorted features to the latent space. As an example after the first feature ($\mu_1$) is sorted per-particle, it is mapped to the event-level

representation as

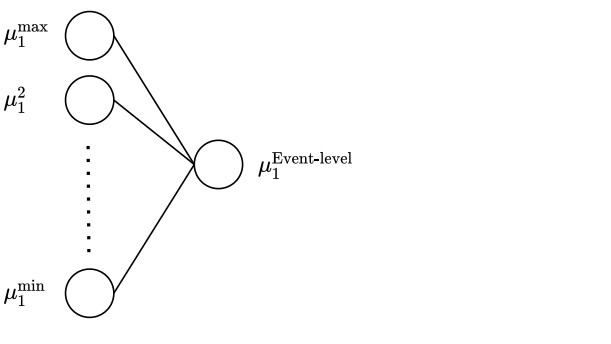

$$,\tag{2}$$

where $\mu_1^{\max(\min)}$ is the maximum (minimum) over the particles. This layer/operation is known as featurewise sort pooling (FSPool) [63].

After the FSPooling layer, we reparameterize as a variational autoencoder with a Gaussian prior. The event-level standard deviation is given as

$$\sigma_1 \to e^{\frac{\sigma_1}{2}}.\tag{3}$$

Then a random representation is drawn using

$$z_1 = \mathcal{G}(\mu_1, \sigma_1),\tag{4}$$

where $\mathcal{G}$ is a Gaussian.

A decoding network is then used to transform the latent data back to a set of four-vectors and particle IDs. We do so using a dense neural network as shown in Fig. 2. Two layers with 256 nodes with ReLU activations are used in the network. The network then splits into two parts. In the first part, there are 80 nodes, representing 20 possible four-vectors. The second part is a series of 180 nodes representing the probability that each of the 20 particles belongs to one of the eight particle IDs or the probability that the four vector should not be included as member of the final set.

Figure 3 shows an illustrative example representation of an input set (left) and encoded-decoded set (right). Each panel is a two dimensional slice across the four-vector representation ($\log_{10} E$, $\log_{10} p_T$, $\eta$, $\phi$) and the points denote the ojects. The color of the points denotes the type of the object. We include the missing transverse momentum as an "object" with zero energy and no $\eta$ information. In the output set, we color the objects based on the highest classification score. This leads to sets that do not make sense from a physics perspective (there is more than one missing transverse momentum object). While this is just a single example, we find that most of the reconstructed sets have similar problem, with too many output particles and missing energy objects in the set. If we were worried about event generation, rather than anomaly detection, procedures could be implemented, such as having a specific output for the missing transverse momentum rather than including it as generic object, or only including objects if the classification score is above some threshold. While the output sets may not look physical, we find that we are still able to achieve decent anomaly detection.

## 3.2 Training

To train the deep set variational autoencoder, the output four-vectors and class predictions are compared with the input set of four-vectors and particle IDs. This is challenging for sets on full events for two reasons. The first is that by construction, the input and output are permutation invariant so we cannot just compare the first element of the input to the first element of the output. One solution to this challenge would be to use a metric such as the Energy Mover's

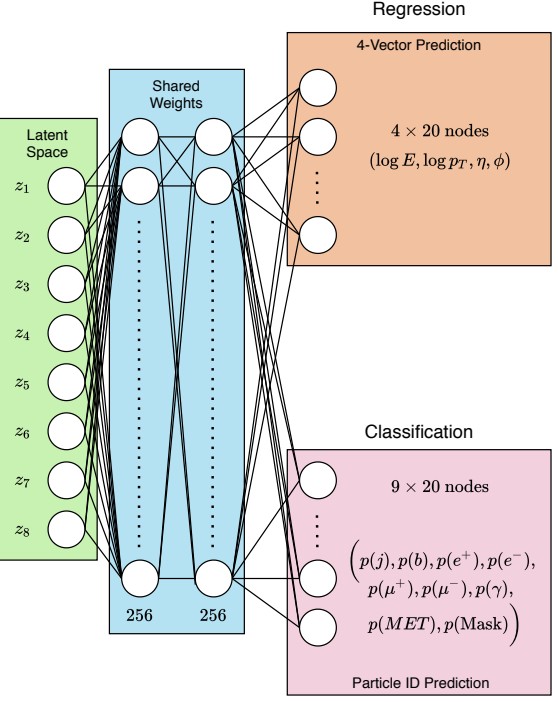

Figure 2: The decoder takes latent space representations of events and outputs a set of particles. The blue box denotes shared wights between the two final output components. The orange box is a set of four-vectors for each of the predicted particles. The pink box represents the classification prediction which assigns a probability to each particle to belong to a certain class or to be masked out.

distance [64] which is based on optimal transport and looks at how much energy is needed to reshuffle one event to look like a different event. However, this brings about the second challenge, what is the cost to change particle types?

With no clear answer to this, we instead implement a modified version of the Chamfer loss. For each particle in the input set, we find the Euclidian distance (in the space of $(\log_{10} E, \log_{10} p_T, \eta, \phi)$) to the closest particle in the output set. To this distance, we then add the log likelihood that the particle in the output is classified as the same particle type. For this, we use the log of the softmax of the output of the classifier. At this stage, if there are any particles in the output set that are not close to any of the input particles, they may not be included in the loss because only the closest particles are used. To account for this, the Chamfer loss then repeats the procedure for every particle in the output set, finding the closest particle in the input set. We again supplement this distance by adding by the probability that they are the same class. It is unclear how to weight the relative importance of the four-vectors being close compared the particle ID being assigned correctly, therefore, we include an extra hyper-parameter, $w$ which could be optimized over. Our modified Chamfer loss is then given by

$$
\begin{aligned}
L_C = \sum_{x \in S_{\text{input}}} & \left( \min_{y \in S_{\text{output}}} \left| \vec{x} - \vec{y} \right| \right) + w \log p(x_{\text{ID}} = y_{\text{ID}}^{\min}) \\
+ \sum_{y \in S_{\text{output}}} & \left( \min_{x \in S_{\text{input}}} \left| \vec{x} - \vec{y} \right| \right) + w \log p(x_{\text{ID}}^{\min} = y_{\text{ID}}).
\end{aligned}
\tag{5}
$$

The last component of the loss for the variational autoencoder enforces structure in the latent space. This is done using the reparameterization trick and computing the KL divergence

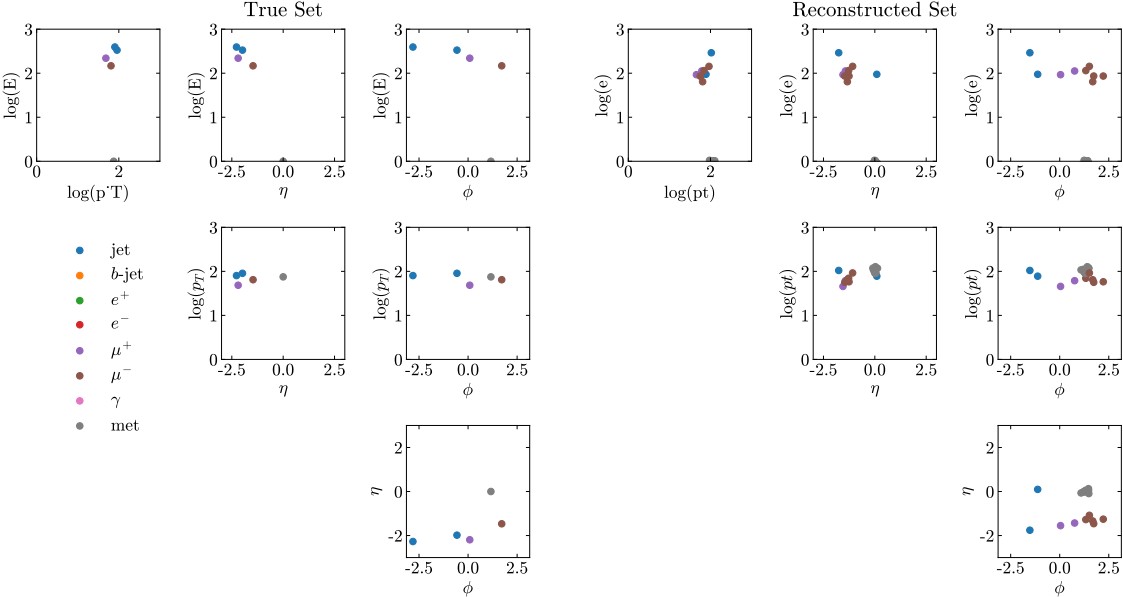

Figure 3: Illustrative example of the input and output structure of the network. The left panels show the original event in different two dimensional slices. The color of each point denotes the type of object. This event is from Channel 2b and contains two jets, a muon pair, and missing transverse momentum. The right panels contain the encoded-decoded set, and there are more objects than in the original set.

between the latent space and a Gaussian prior [65, 66]. This term helps to regularize the network and put similar events nearby in latent space. However, it is unclear how much regularization should be included. Therefore, we include the hyper-parameter $\beta$ to control the relative importance of the KL term and the chamfer loss as

$$L_{\text{total}} = \beta \times KL + \left(1 - \beta\right) \times L_C \,. \tag{6}$$

The network is implemented in PYTORCH [67] and trained using the Adam optimizer [68]. We use the default parameters for Adam, but lower the learning rate by a factor of 10 if the loss computed on the validation set has not improved for 2 epochs. Training ends when the validation loss has not improved for 4 epochs, and typically takes around 30 epochs. We use a batch size of 500 events. We note that the loops required in computing the Chamfer loss makes training quite slow. Because of this, we were unable to explore a huge range of hyper-parameter options. Despite this, our deep set autoencoders were among the best models in the Dark Machines Challenge [50].

# 4 Results

In this work, we experiment with different values of the regularization, $\beta$, and the importance of the classification step of the decoder, $w$. We scan over

$$\begin{aligned} &\beta \in \left\{10^{-6}, 0.001, 0.1, 0.5, 0.8, 0.999, 1.0\right\} \text{ and} \\ &w \in \{1.0, 10.0, 100.0\} \,. \end{aligned} \tag{7}$$

The number of events in channel 3 is so large that training the network was prohibitively slow, and could not be performed for the entire scan (see Sec. 2). Instead, we treat channels 1, 2a,

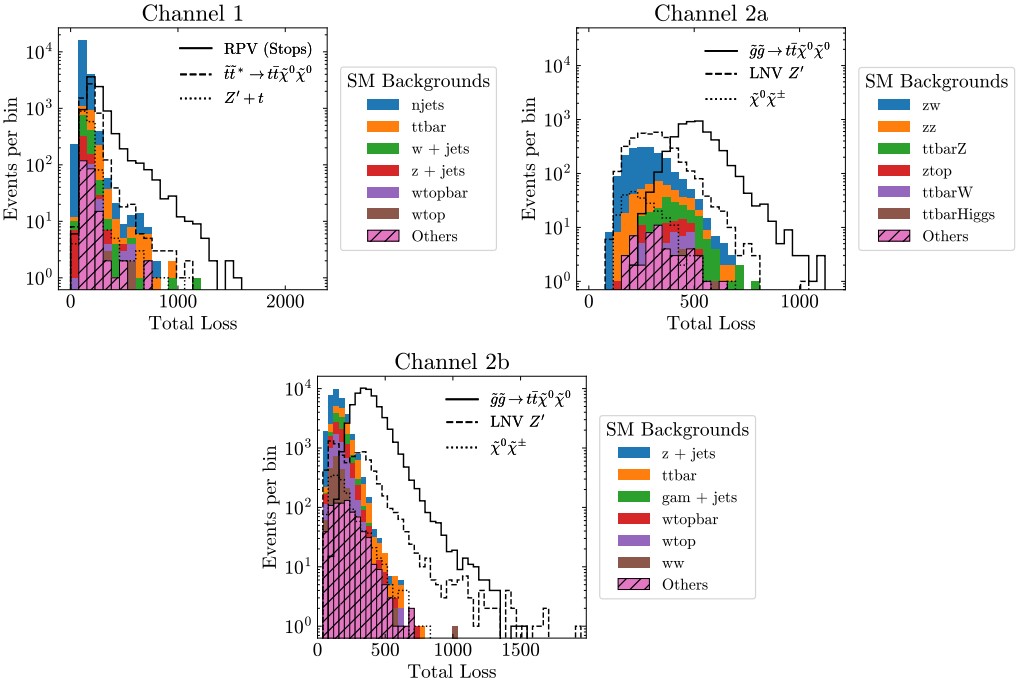

Figure 4: Anomaly score for the networks using $\beta = 10^{-3}$ and $w = 10.0$. The value of $\beta$ determines the ratio of contributions to the loss coming from the set reconstruction and the latent space representation and $w$ controls the relative importance of the four-vector reconstruction versus the particle identification. The networks are trained on SM background events and then applied to a new set of background and BSM events. The background histograms are stacked, while the BSM signals are independently overlaid. Even though the number of dominant background changes for each channel, the same network architecture can be applied to each to separate some signals from the background.

and 2b as open data sets to optimize the values of $\beta$ and $w$ and use channel 3 as a final check to test for generalization.

## 4.1 Explorations on channels 1 and 2

We use the full loss term, including the KL divergence, as our event-by-event anomaly score. This is found to yield good separation between the SM background and many of the signals. For example, Fig. 4 shows the total loss of events from the test set for $\beta = 0.001$ and $w = 10.0$. The colored bars are stacked histograms for the SM background as indicated by the legends. Each channel has different dominant backgrounds. Overlayed on the plots are histograms for three different signals, denoted by the solid, dashed, and dotted black lines. In channel 1, these are RPV stop production, R parity conserving stop production, and a mono-top $Z'$ model, respectively. For channel 2 (both and A and B) we display signal events for R parity conserving gluino pair production, a lepton number violating $Z'$, and R parity conserving chargino-neutralino production. Even though the sets are not reconstructed well (for example see Fig. 3), we see that some of the new physics events have larger loss values than the SM events. In each channel, there is at least one BSM model which is easy to separate from the background, the RPV stops for channel 1 and gluino production for channel 2. However, we also note that there are also BSM models that the network has a hard time distinguishing from the SM, for instance the mono-top $Z'$ for channel 1 and the chargino-neutralino production

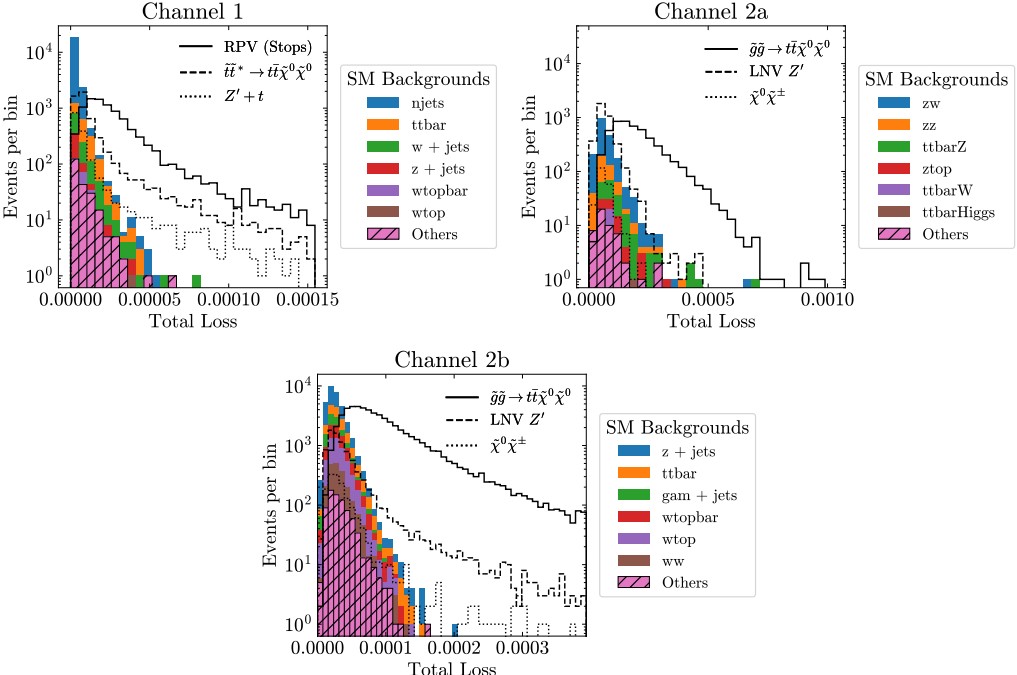

Figure 5: Anomaly score for the networks using $\beta = 1.0$ and $w = 1.0$. The network does not try to reconstruct the set, but only uses the KL divergence in the latent space. This leads to better signal separation than the networks which use a decoder.

for channel 2.

In Fig. 5, we show the similar loss histograms for the network using $\beta = 1.0$ and $w = 1.0$. This value for $\beta$ essentially removes the decoder from the network, and the network is only trying to reduce the set to an eight dimensional Gaussian. The same BSM models are chosen as in Fig. 4 to aid in comparison. In Channel 1, we see that both the $\tilde{t}\tilde{t}^* \to t\bar{t}\tilde{\chi}^0\tilde{\chi}^0$ signal and the $Z' + t$ signal are much more separated from the background when the network does not have the decoder. A similar pattern is observed in the channels 2, with more of the hard to discover signals getting separated from the background.

With many unique BSM signal-Channel combinations, determining the best values of $\beta$ and $w$ is not trivial. Sometimes networks find certain signals better than others, but changing the parameters can trade which signals can be separated from the background. We incorporate a holistic view of the problem as done in the challenge [50, 69]. For each channel, we find the total loss which allows 1%, 0.1%, or 0.01% of the background events through. We then determine the amount of each signal that passes each of these loss thresholds to compute the significance improvement, given by

$$\text{Significance Improvement} = \epsilon_S / \sqrt{\epsilon_B}, \tag{8}$$

where $\epsilon_S$ is the signal efficiency and $\epsilon_B$ is the background efficiency. For each signal in each channel, we keep the threshold which maximizes the significance improvement. In some cases, the looser cut allows through enough signal to be better than a smaller cut, while other times the tighter cut is necessary to remove a larger portion of the background. Some of the BSM physics models have signals which would appear in multiple channels. When this is the case, we denote the total significance improvement (TI) as the largest significance improvement across the channels for that signal.

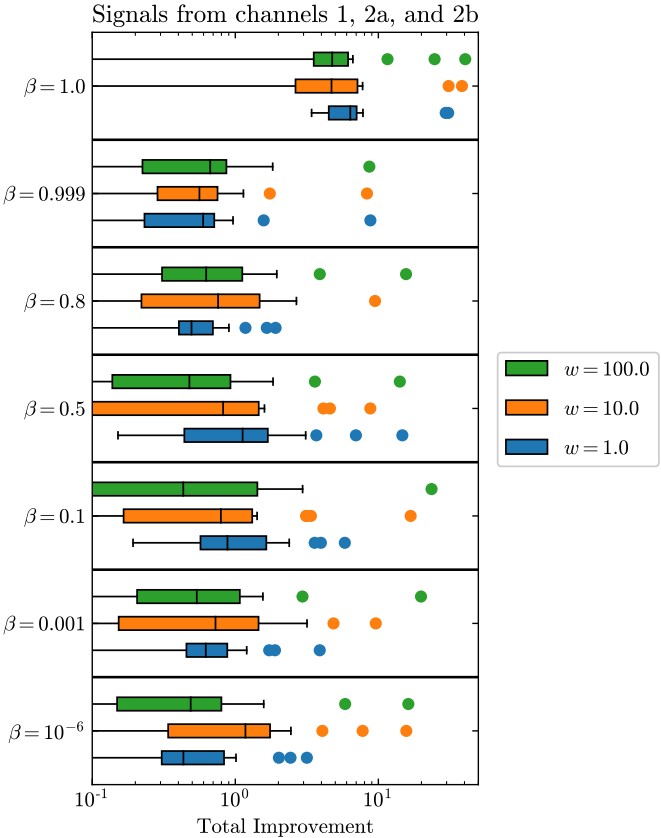

Figure 6: Distribution of total improvement scores from the signals of channels 1, 2a, and 2b. Changing the value of *w* (denoted by the colors) has a minor impact. In contrast, setting $\beta = 0$ removes the decoder and yields significantly better anomaly detection performance.

Figure 6 shows the distribution of the TI scores across the 11 BSM models (18 total mass spectra) for the different networks. The color of the marker denotes the value of *w* and value of $\beta$ is denoted by the region along the *y*-axis. The boxes cover the 25th-75th quantile, with the line denoting the median of the data. The line extend out to either the maximum or minimum, or to 1.5 times the box width from the end of the box, in which case the remaining data are shown as (outlier) points. The minimum TI score is less than 1 for all models, which implies that there is some BSM model which all of the networks cannot distinguish from the SM background.

There appears to be a clear distinction between the anomaly detectors that use the decoder ($\beta \neq 1$) and the method without the decoder ($\beta = 1$). The distribution of TI scores are much larger for the model without the decoder. For instance, the maximum scores are around a factor of 3-4 better, and the median scores are nearly an order of magnitude better. The cause of this is unknown and a subject of future research. Going forward, we will only study the model with $\beta = 1$ in this work.

## 4.2 Application to Channel 3

After determining that the networks which only compress to a small latent space have larger median and maximum total improvements on channels 1, 2a, and 2b, we train these models on channel 3. The resulting significance improvements are shown for each BSM physics signal in Fig. 7. The circle and square markers denote the network trained with $w = 1$ and $w = 10$,



Figure 7: Significance improvements of the BSM signals for the models using $\beta = 1.0$. The maximum significance improvement over three anomaly score cuts are displayed for each channel. The circle and squares mark the networks with $w = 1.0$ and $w = 10.0$, respectively. However, $w$ does not enter into the loss when $\beta = 1$, so the spread between the results can be viewed as a training uncertainty.

respectively. However, the parameter $w$ should not affect the network when $\beta = 1$, so the differences indicate the level of training uncertainty. The blue, orange, green, and red markers show the results for channel 1, 2a, 2b, and 3, respectively. While channel 3 has the most training data, this did not lead to overall better anomaly detection. In only one BSM signal (monojet + $Z' \rightarrow$ dark matter) is the significance improvement greater for channel 3 than the other channels.

Overall, we find that the networks are able to detect most of the BSM models. We note that the performance is worst for channel 2a, which was expected because the training set is quite small. Given this, the networks are not sensitive to the two supersymmetric chargino-neutralino production signals which only appear in channel 2a. All of the other signals which appear in channel 2a also appear in 2b, and the significance improvement is between 5-10.

We find that the best significance improvement tends to come from channel 2b, which is the two lepton channel. At this stage, it is unclear if the performance is better because of the size of the training data, or if the BSM signals that appear in this channel just happen to be easier to detect. Investigating this is left for future work.

## 5   Discussion

We have developed an anomaly detection method for use at the LHC as a model agnostic BSM search. The method uses a variational autoencoder setup, where the input data is compressed to a small latent space and then decompressed and compared with the original input. We use a deep set input data architecture, where each reconstructed particle is viewed as an independent point in the set, and the operations are permutation invariant.

In comparing the input and output sets, we include a term to account for the particle type, and scan over how much to weight this term in comparison the the four momentum reconstruction. In general, we find that placing more emphasis on the particle identification leads to better anomaly detection performance. This is especially important if one wants to use the deep set VAE as a generative model.

Surprisingly, we find that the network trained without a decoder provides the best anomaly detection. This is unintuitive because the network is only trying to compress the data, and it does not need to do so in a way that allows for good decompression. With no decoder or decompression, we can say that the network is mapping the input data to a fixed representation (a multidimensional Gaussian). In the Dark Machines challenge [50], these deep set VAE models were among the top models, achieving median total improvements greater than 2 in both the hackathon datasets (results from this work) and in the secret dataset. Interestingly, the other set of models which had high median scores for both the hackathon and secret data sets also used as part of their anomaly score a mapping to a fixed representation [51]. The fixed representations were either also a multidimensional Gaussian distribution or a fixed vector. These type of networks were called fixed target methods in the Dark Machines Challenge.

In this work, and the Dark Machines challenge as a whole, the anomaly detection models were trained on simulation and applied to simulation using the same underlying parameters. The models which do not use a reconstruction loss, but solely focus on the compression of data to the latent space may be harder to validate when applying to real LHC data. To our knowledge, these fixed target methods have not been applied to single object studies, such as anomalous jet tagging, it would be interesting to see if the fixed target aspect also works well there, or if the methods are better for full event analyses. These questions are left for future study.

## Acknowledgements

We thank Baptiste Ravina and Nathan Simpson for early collaboration on this work. We thank Katie Fraser, Samuel Homiller, Rashmish Mishra, and Sascha Caron for insightful comments on a previous version of this paper. The work of B.O. is supported by the U.S. Department of Energy under contracts DE-SC0013607 and DE-SC0020223. This work is supported by the National Science Foundation under Cooperative Agreement PHY-2019786 (The NSF AI Institute for Artificial Intelligence and Fundamental Interactions, http://iaifi.org/). Computations in this paper were run on the FASRC Cannon cluster supported by the FAS Division of Science Research Computing Group at Harvard University.

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
