# Peer review of "Deep Set Auto Encoders for Anomaly Detection in Particle Physics"

_SciPost Physics, doi:SciPost Phys. 12, 045 (2022)_

## Round 1 · Referee Report · Anonymous (Referee 1) · 2021-10-18

Strengths

  1. Definitely topical, utilization of AEs and latent space for anomaly detection is of significant interest
  2. Clear description of the methodology, datasets, architecture used
  3. Presents multiple interesting results and avenues for future research

Weaknesses

  1. The conclusions seem to largely suggest that the decoding portion of the network is unnecessary for the purposes of anomaly detection, but significant space in the paper is still devoted to discussing the design and tuning of this portion of the network. I think some modification to the presentation of the network design and results could make the paper even more compelling.
  2. Data presentation in plots can be difficult to understand
  3. Some grammatical errors, undefined jargon

Report

This paper presents a new network architecture for anomaly detection, and presents a surprising result that the decoder portion is largely unnecessary. I think that with some minor revisions for clarity/presentation this paper should be published, and some larger revisions would further strengthen the findings and their impact.

Requested changes

  1. Some of the plots are difficult to interpret (specifically Fig. 4 and 5 - suggest combining the rarer backgrounds into one histogram).
  2. Jargon and grammatical errors should be fixed.
  3. Since the decoder portion is concluded to be unnecessary for anomaly detection, I suggest either a clear separation between the results with and without $\beta=1$ or a discussion of their relative performance on certain signals. I would suggest possibly also expanding the results in Fig. 6 to show the full set of TI values for a set of models (both with and without $\beta=1$), since anomaly detection means we do not know which model is out there and thus the distribution across a range of models is more valuable than the min/median/max.

---

## Round 2 · Referee Report · Anonymous (Referee 1) · 2021-11-30

Report

Thank you for the revisions and responses. I think the paper reads quite nicely now and the interesting observation you make about the latent space and decoder is now at the forefront of the discussion. I have no other comments at this time.

---

## Round 2 · Author Response

We thank the referee for their thoughtful comments. We address their comments below.

Referee request: Some of the plots are difficult to interpret (specifically Fig. 4 and 5 - suggest combining the rarer backgrounds into one histogram).
Our Response: Thank you for this recommendation. We have changed the figures to include a much smaller subset of the backgrounds, with the remaining ones marked as “Other”

Referee request: Jargon and grammatical errors should be fixed.
Our Response: many terms have been defined and grammar has been fixed. Glad to correct any more specific instances.

Referee request: Since the decoder portion is concluded to be unnecessary for anomaly detection, I suggest either a clear separation between the results with and without β=1 or a discussion of their relative performance on certain signals. I would suggest possibly also expanding the results in Fig. 6 to show the full set of TI values for a set of models (both with and without β =1) since anomaly detection means we do not know which model is out there and thus the distribution across a range of models is more valuable than the min/median/max.
Our Response: We have tried to make the results with and without the decoder more clear. In making Fig. 6 now show the full distribution, some of the trends we previously discussed about the method with the decoder are more challenging to see. Therefore, we removed those discussions, and now the results basically show that the version without the decoder is much better, so then we only discuss that. We still keep all of the information about the decoder and the discussion about it in the methods section, because they could be relevant for related studies, and it was a part of this experiment.

---

## Round 2 · List of Changes

Added citation to introduction for similar work.
Figures 4 and 5 now show fewer of the background individually to improve readability.
Figure 6 now shows full distribution rather than min, median, max only. Changed some discussion around the figure.

---

## Editorial Decision

published